# The Evolution of Electron Dispersion in the Series of Rare-Earth Tritelluride Compounds Obtained from Their Charge-Density-Wave Properties and Susceptibility Calculations

**DOI:** 10.3390/ma12142264

**Published:** 2019-07-15

**Authors:** Pavel A. Vorobyev, Pavel D. Grigoriev, Kaushal K. Kesharpu, Vladimir V. Khovaylo

**Affiliations:** 1Department of Low Temperature Physics and Superconductivity, M.V. Lomonosov Moscow State University, Moscow 119991, Russia; 2L.D. Landau Institute for Theoretical Physics, Chernogolovka 142432, Russia; 3Department of Theoretical Physics and Quantum Technologies, National University of Science and Technology “MISiS”, Moscow 119049, Russia; 4P.N. Lebedev Physical Institute of RAS, Moscow 119991, Russia; 5National Research South Ural State University, Chelyabinsk 454080, Russia; 6Department of Functional Nanosystems and High Temperature Materials, National University of Science and Technology “MISiS”, Moscow 119049, Russia

**Keywords:** rare-earth tritelluride, charge density wave, transition temperature, electron dispersion, susceptibility, Fermi surface

## Abstract

We calculated the electron susceptibility of rare-earth tritelluride compounds RTe_3_ as a function of temperature, wave vector, and electron-dispersion parameters. Comparison of the results obtained with the available experimental data on the transition temperature and on the wave vector of a charge-density wave in these compounds allowed us to predict the values and evolution of electron-dispersion parameters with the variation of the atomic number of rare-earth elements (R).

## 1. Introduction

In the last two decades, the rare-earth tritelluride compounds RTe_3_ (R = rare-earth elements) were actively studied, both theoretically [1] and experimentally by various techniques [2,3,4,5,6,7,8,9,10,11,12,13,14,15,16,17]. A very rich electronic phase diagram and the interplay between different types of electron ordering [6,7,8], as well as amazing physical effects in electron transport even at room temperature [14,15,16] stimulated this interest. These compounds undergo a transition to a unidirectional charge-density wave (CDW) state with wave vector $Q_{CDW1}\approx \left(0,0,2/7c^{*}\right)$. The corresponding transition temperature $T_{CDW1}$ decreases with the atomic number of the rare-earth element (R) [6]: $T_{CDW1}$ drops from over 600 K in LaTe_3_ [13] to $T_{CDW1}=244\;K$ in TmTe_3_. However, the CDW energy gap does not completely cover the Fermi surface (FS), as can be seen from the ARPES measurements [3,4,5], and the electronic properties below $T_{CDW1}$ remain metallic with a reduced density of electron states at the Fermi level. In RTe_3_ compounds with the heaviest rare-earth elements, the second CDW emerges [6] with the wave vector $Q_{CDW2}\approx \left(2/7a^{*},0,0\right)$ and the transition temperature $T_{CDW2}$ increasing with the atomic number of the rare-earth element (R) [6] from $T_{CDW2}=52\;K$ in DyTe_3_ to $T_{CDW2}=180\;K$ in TmTe_3_. After the second CDW, the RTe_3_ compounds remain metallic, similarly to NbSe_3_. A third CDW has been proposed [12] from the optical conductivity measurements, but not yet confirmed by the X-ray studies. At lower temperatures, the RTe_3_ compounds become magnetically ordered [7]. In addition to all this, at high pressure, the RTe_3_ compounds become superconducting [8].

To understand the richness of this phase diagram and the physical properties in each phase, it is very helpful to have information about the evolution of electronic structure of RTe_3_ compounds with the change of the atomic number of R. Unfortunately, the ARPES data are available only for very few compounds of this family and, in spite of a notable progress in instrumentation, still have a large error bar. The electron transport measurements are much more sensitive, but they only give indirect information about the electronic structure, because of a large number of electron scattering mechanisms [14,15,16]. Similar to the change of the electron–phonon interaction value from alkali elements to transition elements [18], there is a difference in electronic behavior of rare earth elements. La, Ce, Pr, Nd, Pm, and Sm have a small electron–phonon interaction, while Eu, Gd, Tb, Dy, Ho, and Er have a larger one, effecting the electric conductivity of their oxide compounds [18,19,20,21]. As Te lies in the same row as oxygen, one may expect similar behavior for rare-earth tritellurides. In this paper, we use the extensive experimental data on the evolution of the CDW_1_ wave vector $Q_{CDW1}$ and transition temperature $T_{c}$ to study the evolution of the electronic structure of $RTe_{3}$ compounds. We calculate the electron susceptibility, responsible for CDW_1_ instability, as a function of the wave vector and temperature at various parameters, which determine the electron dispersion. The comparison of the results obtained with available experimental data allows us to make predictions about the evolution of these electron-structure parameters with the atomic number of R.

## 2. Calculation

At temperatures $T>T_{CDW1}$, the in-plane electron dispersion in RTe_3_ is described by a 2D tight binding model of the Te plane as developed in [3], where the square net of Te atoms in each conducting layer forms two orthogonal chains created by the in-plane $p_{x}$ and $p_{z}$ orbitals. Correspondingly, *x* and *z* are the in-plane directions. In this model, *t*_‖_ and *t*_⊥_ are the hopping amplitudes (transfer integrals) parallel and perpendicular to the direction of the considered *p* orbital. The resulting in-plane electron dispersion can be written down as:(1)$$\begin{array}{cc} \varepsilon_{1}\left(k_{x},k_{z}\right)= & -2t_{\|}\cos\left[\left(k_{x}+k_{z}\right)a/2\right]-2t_{⊥}\cos\left[\left(k_{x}-k_{z}\right)a/2\right]-E_{F},\\ \varepsilon_{2}\left(k_{x},k_{z}\right)= & -2t_{\|}\cos\left[\left(k_{x}-k_{z}\right)a/2\right]-2t_{⊥}\cos\left[\left(k_{x}+k_{z}\right)a/2\right]-E_{F}, \end{array}$$
where the calculated parameters for DyTe_3_ are *t*_‖_ = 1.85 eV and *t*_⊥_ = 0.35 eV [3] and the in-plane lattice constant a≈4.305 Å [7]. The Fermi energy $E_{F}$ is determined from the electron density, namely from the condition of 1.25 electrons for each $p_{x}$ and $p_{z}$ orbitals [3]. This condition gives us $E_{F}=-2t_{\|}\cos\left(\pi \left(1-\sqrt{3/8}\right)\right)$. It is slightly (by 10%) less than the originally-used Fermi energy value $E_{F}=-2t_{\|}\sin\left(\pi /8\right)$, inaccurately determined [3] from the same condition. The resulting expression shows the relation between these two parameters $t_{\|}$ and $E_{F}$, which is important because they both affect the electron susceptibility.

For the calculation, we use the Kubo formula for the susceptibility of quantity *A* with respect to quantity *B* (see §126 of [22]):(2)$$\chi \left(\omega \right)=\frac{i}{\hbar }\int_{0}^{\infty }\langle \left[\hat{A}\left(t\right),\hat{B}\left(0\right)\right]\rangle e^{i\omega t}dt.$$

For the free electron gas in the terms of the matrix elements, it becomes:(3)$$\chi \left(\omega \right)=\sum_{ml}A_{ml}B_{lm}\frac{n_{F}\left(E_{m}\right)-n_{F}\left(E_{l}\right)}{E_{l}-E_{m}-\omega -i\delta },$$
where *m* and *l* denote the quantum numbers $\left\{k,s,\alpha \right\}$, which are the electron momentum ***k***, spin *s*, and the electron band index *α*. In the CDW response function, the quantities *A* and *B* are the electron density, so that Equation (Equation 2) is a density-density correlator. To study the CDW onset, one needs the static susceptibility at *ω* = 0, but at a finite wave vector ***Q***. Electron spin only leads to a factor of four in susceptibility, but the summation over band index *α* must be retained if there is more than one band crossing the Fermi level. As a result, we have for the real part of electron susceptibility:(4)$$\chi \left(Q\right)=\sum_{\alpha ,\alpha^{\prime }}\int \frac{4d^{d}k}{{\left(2\pi \right)}^{d}}\frac{n_{F}\left(E_{k,\alpha }\right)-n_{F}\left(E_{k+Q,\alpha^{\prime }}\right)}{E_{k+Q,\alpha^{\prime }}-E_{k,\alpha }},$$
where $n_{F}\left(\varepsilon \right)=1/\left(1+\exp\left[(\varepsilon -E_{F})/T\right]\right)$ is the Fermi–Dirac distribution function and dis the dimension of space. Since the dispersion in the interlayer *y*-direction is very weak in RTe_3_ compounds, we can take *d* = 2. Each of the band indices *α* and $\alpha^{\prime }$ may take any of two values 1, 2, because in RTe_3_ two electron bands cross the Fermi level. Here, we assume that the matrix elements $A_{ml}$ and $B_{lm}$ do not depend on the band index. This means that due to the e–e interaction, the electrons may scatter to any of the two bands with equal amplitudes. This assumption has virtually no effect on both the temperature and ***Q***-vector dependence of the electron susceptibility, because the latter is determined mainly by the diagonal (in the band index) terms, which are enhanced in RTe_3_ by a good nesting.

Using Equation (Equation 4), we calculate the electron susceptibility *χ* as a function of CDW wave vector ***Q*** and temperature for various parameters $t_{\|}$ and $t_{⊥}$ of the bare electron dispersion (Equation 1). The CDW phase transition happens when $\chi \left(Q,T\right)U=1$, where the interaction constant *U* only weakly depends on the rare-earth atom in the RTe_3_ family. The position of susceptibility maximum $\chi \left(Q\right)$ gives the wave vector $Q_{CDW1}$ of CDW instability as a function of the band-structure parameters $t_{\|}$ and $t_{⊥}$. The value of susceptibility in its maximum as a function of temperature $\chi_{max}\left(T\right)$ gives the evolution of CDW transition temperature $T_{CDW1}$ as a function of $t_{\|}$ and $t_{⊥}$.

## 3. Results and Discussion

First, we analyze the evolution of the CDW_1_ wave vector. The experimentally-observed dependence of $Q_{CDW1}$ on the atomic number of R-atom can be taken, e.g., from [13]: $Q_{CDW1}$ monotonically increases by ≈10% with the increase of R-atom number from $Q_{CDW1}\approx 0.275$ reciprocal lattice units (r.l.u.) in LaTe_3_ to $Q_{CDW1}\approx 0.303$ r.l.u. in TmTe_3_. The dependence of the CDW wave vector *c*-component, $Q_{CDW1}=\left(0,0,Q_{CDW1}\right)$, on the perpendicular hopping term $t_{⊥}$, calculated using Equation (Equation 4), is shown in Figure 1. As we can see from this graph, $Q_{CDW1}\left(t_{⊥}\right)$ demonstrates approximately linear dependence. The value $t_{⊥}$ = 0.35 eV, proposed in [3] from the band structure calculations, is located in the middle of this plot. The obtained $Q_{CDW1}\left(t_{⊥}\right)$ dependence was rather weak: while $t_{⊥}$ increased dramatically, from 0.2–0.5 eV, and $Q_{CDW1}$ changed by only ∼8% in ${Å}^{-1}$. In the reciprocal lattice units (r.l.u.), this variation was slightly stronger, as the lattice constant *c* decreased with the atomic number from c≈4.407 Å in LaTe_3_ to c≈4.28 Å in ErTe_3_ and TmTe_3_, and the r.l.u. correspondingly increased in ${Å}^{-1}$. However, just the $Q_{CDW1}\left(t_{⊥}\right)$ dependence cannot explain the observed evolution of the CDW_1_ wave vector with the R-atom number, because it is too weak.

The dependence *χ*($t_{⊥}$) is shown in Figure 2. The electron susceptibility varied within one percent of its maximum value and thus remained almost constant. The $\chi_{CDW1}$ values were calculated on the wave vectors $Q_{CDW1}$, obtained for each value of $t_{⊥}$ as a position of the susceptibility maximum. From this plot, we conclude that the parameter $t_{⊥}$ had almost no effect on the CDW_1_ transition temperature. Hence, to interpret the evolution of CDW_1_ transition temperature $T_{CDW1}$ and of its wave vector $Q_{CDW1}$ with the rare-earth atomic number, one needs to consider their $t_{\|}$-dependence.

The dependence $Q_{CDW1}\left(t_{\|}\right)$ is shown in Figure 3. The interval of this plot comprises the values $t_{\|}=1.7$ eV and $t_{\|}=1.9$ eV, obtained in [3] from the band structure calculations for the lightest and heaviest rare-earth elements. $Q_{CDW1}$($t_{\|}$) demonstrated sublinear monotonic dependence, but $Q_{CDW1}$ increased with the increasing of parameter $t_{\|}$. This was opposite to the dependence $Q_{CDW1}\left(t_{⊥}\right)$. Comparing Figure 3 with the experimental data on $Q_{CDW1}$, summarized in [13], we may conclude that the parameter $t_{\|}$ increased with the atomic number of the rare-earth element. According to the band structure calculations in [3], this transfer integral indeed increased from $t_{\|}=1.7$ eV in LaTe_3_ to $t_{\|}=1.9$ eV in LuTe_3_. Thus, our conclusion qualitatively agrees with the band-structure calculations in [3]. However, according to our calculation, the variation of $t_{\|}$ with the atomic number of the rare-earth element must be stronger in order to account for the observed $Q_{CDW1}$ dependence.

In Figure 4, we plot the calculated $\chi (t_{\|})$ dependence, which was approximately linear. Similar to our calculations of $\chi (t_{⊥})$, the susceptibility value was taken in its maximum as a function of the wave vector $Q_{CDW1}$. *χ* changed significantly: about 35% of its maximum value in the full range of parameter $t_{\|}$ change. The ${CDW}_{1}$ transition temperature $T_{c}$ is given by the equation [23] $|U\chi (Q_{CDW1},T_{c})|=1$. Since the susceptibility increased with the decrease of temperature, the largest value of *χ* corresponded to the highest value of CDW transition temperature. We assumed that the electron–electron interaction constant *U* remained almost the same for the considered series of RTe_3_ compounds, because they have a very close electronic structure. The result obtained (see Figure 4) was comparable to the change of transition temperature $T_{CDW1}$ observed in the RTe_3_ series [7]. The value $t_{\|}=1.85$ eV in $DyTe_{3}$ was the reference point. The experimentally-observed transition temperature to the $CDW_{1}$ state in TmTe_3_ was $T_{CDW1}=245$ K, while for GdTe_3_, it was $T_{CDW1}=380$ K and for DyTe_3_
$T_{CDW1}=302$ K [7]. This transition temperature was reduced by 35% of its maximum value from GdTe_3_ to TmTe_3_. Thus, we may assume that this range of $t_{\|}$ described the whole series of compounds from TmTe_3_ to GdTe_3_. Moreover, basing on our calculations, we predicted the values $t_{\|}\approx 1.37$ eV in GdTe_3_, $t_{\|}\approx 1.96$ eV in HoTe_3_, $t_{\|}\approx 2.06$ eV in ErTe_3_, and $t_{\|}\approx 2.20$ eV in TmTe_3_.

The dependence $\chi (t_{⊥})$ calculated at temperatures above the transition is shown in Figure 5. It is important to note there that with the decrease of temperature, the wave vector did not shift and thus did not change its value, as shown in Figure 6: the position of the maximum of susceptibility was almost the same for two different temperatures. Thus, the electronic susceptibility in Figure 5 was calculated on the same $Q_{max}$ wave vectors in Figure 1, but had a lower value with the increase of temperature from 240 K–400 K.

The transition temperatures and conducting band parameters for various RTe_3_ compounds are summarized in Table 1. $t_{\|}$ increased with the increase of the atomic number of R. The observed evolution of $Q_{max}\left(t_{⊥}\right)$ suggests that *t*_⊥_ decreased with the increase of the atomic number of R, but since the electronic susceptibility was almost independent of *t*_⊥_, we could not predict the *t*_⊥_ values.

Our suggested values of the transfer integrals *t*_‖_ and *t*_⊥_ assumed that (1) the effective electron–electron interaction at the CDW wave vector did not depend considerably on the R, and (2) the condition of 1.25 electrons for each $p_{x}$ and $p_{z}$ orbitals was fulfilled for all studied RTe_3_ compounds.

## 4. Conclusions

To summarize, we calculated the electron susceptibility on the CDW_1_ wave vector in the rare-earth tritelluride compounds as a function of temperature, wave vector, and two tight-binding parameters (*t*_∥_ and *t*_⊥_) of the electron dispersion. From these calculations, we showed that the parameter *t*_⊥_ had almost no effect on the CDW_1_ transition temperature $T_{CDW1}$ and weakly affected the CDW_1_ wave vector $Q_{CDW1}$. On the contrary, the variation of parameter *t*_∥_ with the atomic number *n* of rare-earth element drove the variation of both $T_{CDW1}$ and $Q_{CDW1}$. Note that the increase of *t*_‖_ and of *t*_⊥_ had opposite effects on $Q_{CDW1}$. Using the experimentally-measured transition temperatures $T_{CDW1}$, we estimated the values of *t*_‖_ from our calculations for the whole series of RTe_3_ compounds from TmTe_3_ to GdTe_3_.

## Figures and Tables

**Figure 1 materials-12-02264-f001:**
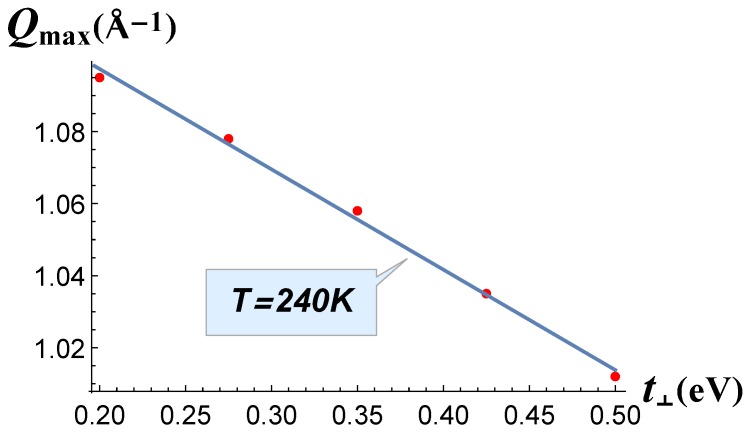
CDW_1_ (charge-density wave) vector $Q_{max}$ calculated at *T* = 240 K as a function of the electron hopping term *t*_⊥_.

**Figure 2 materials-12-02264-f002:**
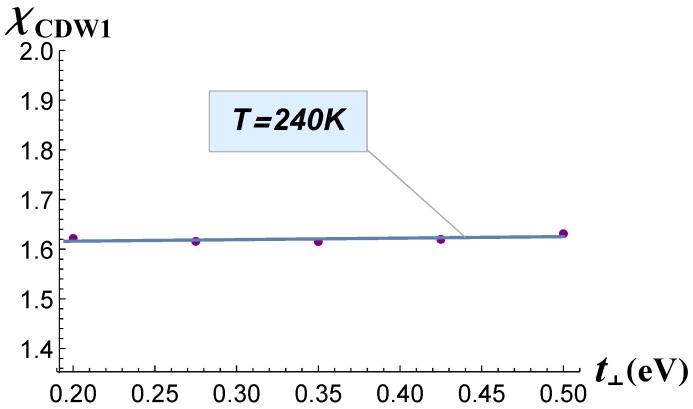
Electron susceptibility *χ* calculated at *T* = 240 K as a function of the electron hopping term *t*_⊥_.

**Figure 3 materials-12-02264-f003:**
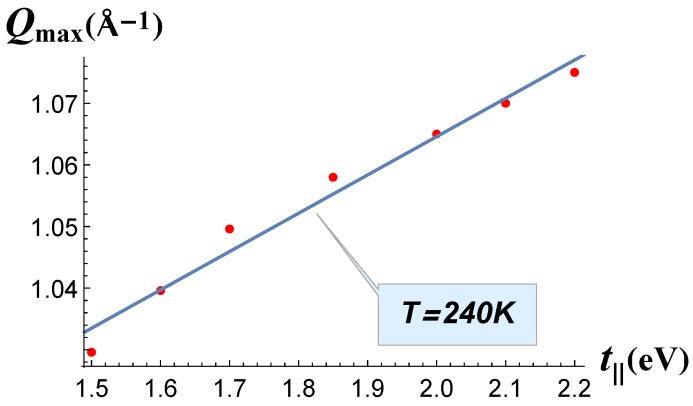
CDW_1_ wave vector $Q_{max}$ calculated at *T* = 240 K as a function of the electron hopping term *t*_∥_.

**Figure 4 materials-12-02264-f004:**
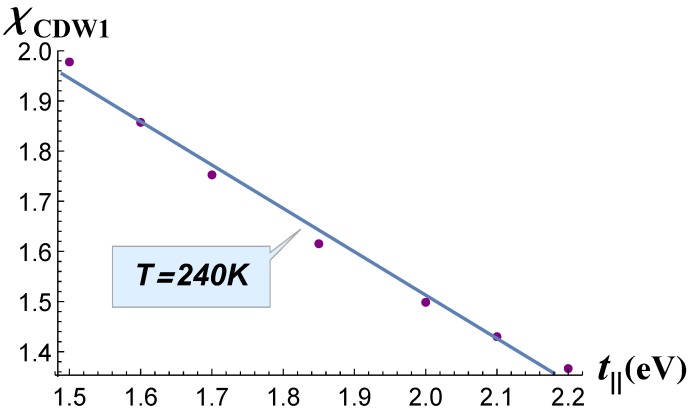
Electron susceptibility *χ* calculated at *T* = 240 K as a function of the electron hopping term *t*_∥_.

**Figure 5 materials-12-02264-f005:**
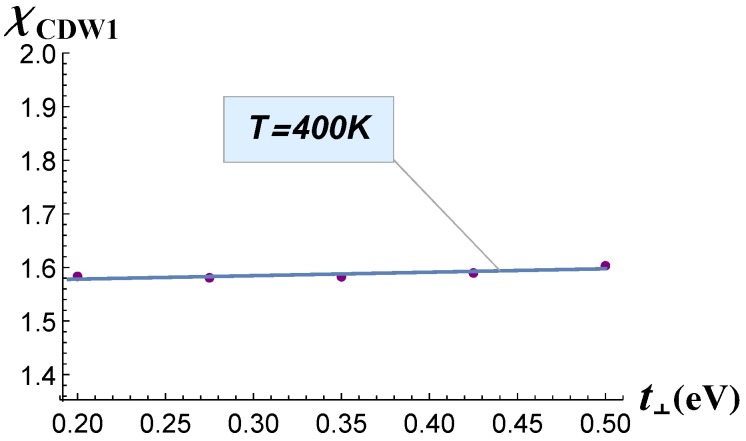
Electron susceptibility *χ* calculated at *T* = 400 K as a function of the electron hopping term *t*_⊥_.

**Figure 6 materials-12-02264-f006:**
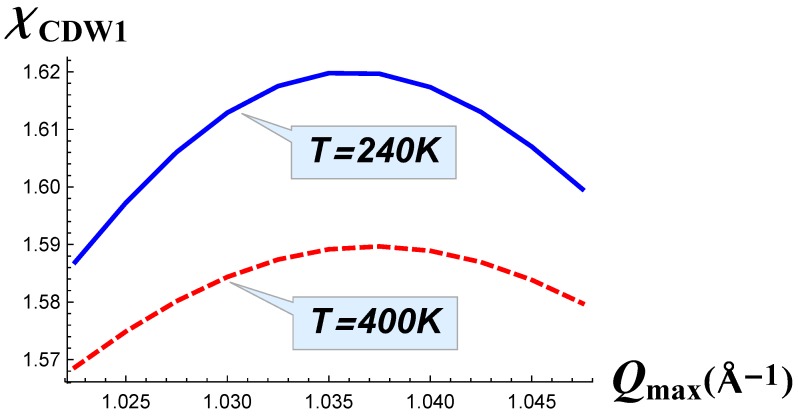
The total susceptibility as a function of wave vector $Q_{max}$ near its maximum calculated at *T* = 240 K (solid blue line) and at *T* = 400 K (dashed red line).

**Table 1 materials-12-02264-t001:** List of parameters describing the dispersion and the CDW transition temperatures $T_{CDW2}$ and $T_{CDW2}$ for rare-earth elements (R) Gd, Dy, Ho, Er, and Tm.

Compound	$T_{CDW1}$ [6], K	$T_{CDW2}$ [6], K	Lattice Parameter [6], Å	$t_{∥}$, eV	$t_{⊥}$, eV	$E_{F}$, eV
GdTe_3_	377	-	4.320	≈1.37	>0.35	0.95
DyTe_3_	306	49	4.302	1.85 [3]	0.35 [3]	1.28
HoTe_3_	284	126	4.290	1.96	<0.35	1.35
ErTe_3_	267	159	4.285	2.06	<0.35	1.42
TmTe_3_	244	186	4.275	2.20	<0.35	1.52

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
