# Peer review of "The Evolution of Electron Dispersion in the Series of Rare-Earth Tritelluride Compounds Obtained from Their Charge-Density-Wave Properties and Susceptibility Calculations"

_materials, 2019, doi:10.3390/ma12142264_

Reviewer 1 Report

The manuscript reports on the calculation of electron susceptibility of rare-earth tritelluride compounds (from Tm to Gd) as a function of temp, wave vector and electron-dispersion parameters. The rationale for the work is clearly defined and the manuscript is well presented. Hence, it can be considered for publication in Materials after a minor revision.

Few minor comments and queries:

1.     Page 2 – The authors claim in the introductory section that due to a large number of electron scattering mechanisms, the electron transport measurements are indirect. What do they mean by that? How where those measurements made (ZEM / PPMS)? They have used these experimental data to study the evolution of the electronic structure of RTe3 compounds, but no background information pertaining to those experimental data was given.

2.     Page 3 (lines 72 – 74) - Can the authors point out some reasons why just the Q dependence cannot explain the observed evolution of the wave vector with the R-atom number?

3.     On a general note, the authors haven’t mentioned how their calculation is comparable with other such similar studies that were made earlier (from the literature). How reliable are their estimation (if possible, give the % of error or standard deviation)?

Author Response

We thank the referee for careful reading of our manuscript and for his comments.

Referee comment:

The manuscript reports on the calculation of electron susceptibility of rare-earth tritelluride compounds (from Tm to Gd) as a function of temp, wave vector and electron-dispersion parameters. The rationale for the work is clearly defined and the manuscript is well presented. Hence, it can be considered for publication in Materials after a minor revision.

Few minor comments and queries:

1.     Page 2 – The authors claim in the introductory section that due to a large number of electron scattering mechanisms, the electron transport measurements are indirect. What do they mean by that? How where those measurements made (ZEM / PPMS)? They have used these experimental data to study the evolution of the electronic structure of RTe3 compounds, but no background information pertaining to those experimental data was given.

Our reply:

By the phrase “the electron transport measurements are indirect” we didn’t mean that the conductivity is measured indirectly, but only that they give indirect information about the electronic structure. Of course, the cited conductivity measurements [15-17] are direct and accurate, and their method is described in detail in these references.

To remove this ambiguity, in the revised version we rephrase this as “they only give indirect information about the electronic structure”.

Referee comment

2.     Page 3 (lines 72 – 74) - Can the authors point out some reasons why just the Q dependence cannot explain the observed evolution of the wave vector with the R-atom number?

Our reply:

One needs QCDW1(t||) dependence, because QCDW1(t) dependence is rather weak, even on the large interval of t variation. The QCDW1(t||) dependence is much stronger.

To make it clearly, in the revised version we add the phrase “because it is too weak”.

Referee comment:

 3.     On a general note, the authors haven’t mentioned how their calculation is comparable with other such similar studies that were made earlier (from the literature). How reliable are their estimation (if possible, give the % of error or standard deviation)?

Our reply:

There are no such previous calculations to compare with our results. The values of the transfer integrals, presented in Ref.1 and Ref.4, we use in our calculations as reference points. As an example, the authors of Ref. 4 have obtained the transfer integral values by reproducing the calculated density of states at Fermi level and the band structure. In our work the trend of changing the transfer integral values with the atomic number of rare-earth element R corresponds to this change in Ref.1 and Ref.4. We have compared our results with those in Ref. 4 in lines 86-90 (88-91 in the revised version).

Reviewer 2 Report

This manuscripts describes calculations about the electron susceptibility due to charge-density-waves. The manuscript is not clearly written and hard to understand for the target readers of the magazine “Materials”. The aim of the manuscript is to calculate the electron susceptibility of rare-earth tri-telluride compounds RTe3, which are known for their rich physics. The calculations are only shown for one temperature below room temperature.

1) Line 10: Ref 4 states “electron-phonon coupling is required to stabilize the CDW”. In the authors calculation the values of EPI is not mentioned at all. Please provide the values which you have used and how it contributes. There is a difference in Rare earth elements. Sm, Nd, Pr, La have a totally different behavior than Yb, Er, Ho, Dy Gd, Eu concerning their electronic shell, which affects their Electron-phonon-interaction. This is not mentioned in the introduction.

2) Line 38/45: Is this calculation a pure analytical calculation? Please mention this and describe the calculation method more clearly and/or include a reference where this method has been used already.

3) Line 74: Are Figure 1 to 4 specific for one phase or general, please kindly mention. If you have data for other temperature, please include. Can the figures be extrapolated to higher or lower t-values? What is the limit?

4) Line 105: For better readability, please kindly provide a table with your result, e.g. t_parallel, t_perpendicular, T_CDW, susceptibility and used lattice parameters for all RTe3 which were calculated and if possible compare to experimental results (line 32). Electron susceptibility is correlated to optical conductivity, please try to explain how your resulst effect the refraction index.

5) Line 110: English correction:

…does not almost affect the CDW -> …does almost not affect the CDW -> has almost no effect on the CDW

Author Response

We thank the referee for careful reading of our manuscript and sending us his/her comments.

Referee comment:

This manuscripts describes calculations about the electron susceptibility due to charge-density-waves. The manuscript is not clearly written and hard to understand for the target readers of the magazine “Materials”. The aim of the manuscript is to calculate the electron susceptibility of rare-earth tri-telluride compounds RTe3, which are known for their rich physics. The calculations are only shown for one temperature below room temperature.

Our reply:

In the revised version we have added Figs. 5,6 to show the results of our calculations at several temperatures.

Referee comment:

1) Line 10: Ref 4 states “electron-phonon coupling is required to stabilize the CDW”. In the authors calculation the values of EPI is not mentioned at all. Please provide the values which you have used and how it contributes. There is a difference in Rare earth elements. Sm, Nd, Pr, La have a totally different behavior than Yb, Er, Ho, Dy Gd, Eu concerning their electronic shell, which affects their Electron-phonon-interaction. This is not mentioned in the introduction.

 Our reply:

The electron-phonon interaction (EPI) indeed contributes to CDW formation and its stability by changing (usually, increasing) the effective electron-electron coupling at CDW wave vector. Although this contribution is, usually, smaller than the direct e-e interaction, even a small change of e-e coupling strongly affects the CDW transition temperature. However, we do not calculate the e-e coupling but take it as some unknown constant. Instead we focus on the electron susceptibility which is also responsible for CDW stability and depends much stronger on electron band parameters.

The e-e coupling U, renormalized by EPI, can be estimated semi-phenomenologically from our susceptibility calculations and the know transition temperature, but we don’t have the e-e coupling values from other sources to compare with.  

Concerning the dependence of EPI coupling in RTe_3 on the rare-earth element R, it is unknown, but smaller than EPI coupling itself. We do not consider it in our calculations. To indicate the approximations made, in the end of discussion we add the new phrase: “Our calculations assume that the effective electron-electron interaction at CDW wave vector does not depend considerably on the rare-earth element R.”

Referee comment 2) Line 38/45: Is this calculation a pure analytical calculation? Please mention this and describe the calculation method more clearly and/or include a reference where this method has been used already.

Our reply:

All formulas are derived analytically. Then the integrals in Eq. (4) are calculated numerically to present the explicit results, shown in all figures.

Referee comment 3) Line 74: Are Figure 1 to 4 specific for one phase or general, please kindly mention. If you have data for other temperature, please include. Can the figures be extrapolated to higher or lower t-values? What is the limit?

Our reply:

a)     Yes, these figures 1-4 are for one temperature. In the revised version we have included additional figure 5,6 for a different temperature above the CDW transition.

b)    Eq. (4) is valid for any dispersion and its parameters. The chosen range of t_perp and t_par is sufficient for our purpose, because it corresponds to the experimental range of transition temperature. Further increase of the parameters leads to a stronger deviation from the linear dependence. The FS curvature is proportional to tperp /tpara. If this ratio exceeds unity, the FS topology changes, which leads to additional effects.

Referee comment 4) Line 105: For better readability, please kindly provide a table with your result, e.g. t_parallel, t_perpendicular, T_CDW, susceptibility and used lattice parameters for all RTe3 which were calculated and if possible compare to experimental results (line 32). Electron susceptibility is correlated to optical conductivity, please try to explain how your resulst effect the refraction index.

 Our reply:

a)     The table is added in the revised version.

b)    Indeed, CDW affects the refraction index. The refraction index depends on dielectric constant, which is affected by CDW (see Ref. [19] for details).

5) Line 110: English correction:

…does not almost affect the CDW -> …does almost not affect the CDW -> has almost no effect on the CDW

Our reply: Corrected in the revised version.

Round  2

Reviewer 2 Report

The manuscript has been improved remarkably. Two more figures/tables have been included and one more temperature dependence calculation has been included. The text has been corrected and improved.

For including the mentioned physical behavior of electronic resonance in the sequence of lanthanides, there is the suggestion to add the following sentences at line 32 after “[13-17]” and the corresponding references between present [18] and [19].

Similar as in the change of the electron-phonon-interaction value from alkali elements to transition elements [18], there is a difference in electronic behavior of rare earth elements. La, Ce, Pr, Nd, Pm and Sm have a small electron-phonon-interaction, while Eu, Gd, Tb, Dy, Ho, Er, have a large one, effecting the electric conductivity of their oxide compounds [19-22]. As Te lies in the same row than oxygen, we can expect similar behavior for rare-earth tritellurides.

[19] S. Y. Savrasov, D. Y. Savrasov, Electron-phonon interactions and related physical properties of metals from linear-response theory, Phys. Rev. B 54 (1996) 23 16487-16501

[20] Pieter Dorenbos, The electronic level structure of lanthanide impurities in REPO4, REBO3, REAlO3, and RE2O3 (RE = La, Gd, Y, Lu, Sc) compounds, J. Phys.: Condens. Matter 25 (2013) 225501, doi:10.1088/0953-8984/25/22/225501.

[21] W. Wunderlich, and H. Ohsato, Enhanced Microwave Resonance Properties of Pseudo-Tungsten-Bronze Ba6-3xR8+2xTi18O54 (R= Rare Earth) Solid Solutions Explained by Electron–Phonon Interaction, Journal Jap. Appl. Phys. 52 (2013), doi: 10.7567/JJAP.52.09KH04

[22] Joohwi Lee, Nobuko Ohba and Ryoji Asahi, First-principles prediction of high oxygen-ion conductivity in trilanthanide gallates Ln3GaO6, Sci. Technol. Adv. Mater. 20 (2019) 144-159, doi: 10.1080/14686996.2019.1578183

Author Response

We have added the phrase and new references as the referee suggested.